# Artificial Intelligence in the Management of Women with Endometriosis and Adenomyosis: Can Machines Ever Be Worse Than Humans?

**DOI:** 10.3390/jcm13102950

**Published:** 2024-05-16

**Authors:** Giulia Emily Cetera, Alberto Eugenio Tozzi, Valentina Chiappa, Isabella Castiglioni, Camilla Erminia Maria Merli, Paolo Vercellini

**Affiliations:** 1Gynecology Unit, Fondazione IRCCS Ca’ Granda Ospedale Maggiore Policlinico, 20122 Milan, Italy; giuliaemily.cetera@gmail.com (G.E.C.); direzione.ginecologia@policlinico.mi.it (C.E.M.M.); 2Academic Center for Research on Adenomyosis and Endometriosis, Department of Clinical Sciences and Community Health, Università degli Studi di Milano, 20122 Milan, Italy; 3Predictive and Preventive Medicine Research Unit, Bambino Gesù Children’s Hospital, IRCCS, 00165 Rome, Italy; albertoeugenio.tozzi@opbg.net; 4Gynaecologic Oncology, Fondazione IRCCS Istituto Nazionale dei Tumori, 20133 Milan, Italy; valentina.chiappa@istitutotumori.mi.it; 5Department of Physics, Università Milano Bicocca, 20126 Milan, Italy; isabella.castiglioni@unimib.it

**Keywords:** artificial intelligence, endometriosis, adenomyosis

## Abstract

Artificial intelligence (AI) is experiencing advances and integration in all medical specializations, and this creates excitement but also concerns. This narrative review aims to critically assess the state of the art of AI in the field of endometriosis and adenomyosis. By enabling automation, AI may speed up some routine tasks, decreasing gynecologists’ risk of burnout, as well as enabling them to spend more time interacting with their patients, increasing their efficiency and patients’ perception of being taken care of. Surgery may also benefit from AI, especially through its integration with robotic surgery systems. This may improve the detection of anatomical structures and enhance surgical outcomes by combining intra-operative findings with pre-operative imaging. Not only that, but AI promises to improve the quality of care by facilitating clinical research. Through the introduction of decision-support tools, it can enhance diagnostic assessment; it can also predict treatment effectiveness and side effects, as well as reproductive prognosis and cancer risk. However, concerns exist regarding the fact that good quality data used in tool development and compliance with data sharing guidelines are crucial. Also, professionals are worried AI may render certain specialists obsolete. This said, AI is more likely to become a well-liked team member rather than a usurper.

## 1. Introduction

Back in 1954, Roald Dahl wrote *The Great Automatic Grammatizator*, a story of a young engineer who developed a machine to mass-produce literature. The machine was so successful that the engineer opened a literary agency and bought out real authors, paying them to never write again. At the end of the story, over half of all the novels published in the United Kingdom had been written by the machine and writers who had resisted the temptation of being bought out by the engineer dieds of hunger [1].

In those days, authors feared that the advent of the computer would sabotage artistic writing, in a similar way to how nowadays we are concerned that artificial intelligence (AI) will dehumanize medical care and usurp both physicians and researchers.

However, can AI really be worse than humans? As AI is rapidly being integrated in all medical specializations, time has come to abandon magical thoughts and catastrophic interpretations and to start critically appraising this novel form of technology.

As critical thought stems from knowledge, we will briefly describe what the term AI actually refers to.

A unique definition of AI does not exist. Today, we are witnessing an enormous effort in trying to resolve this issue, in particular due to the implications that AI is having in the availability of innovative solutions on the market. Europe is trying to provide a definition that has legal value with the EU AI ACT, classifying AI technologies according to the risk they pose to the user. For the purpose of this review, we can refer to AI as any machine that can carry out a complex task, which would typically involve biological brainpower (including physical tasks) [2]. Among the variety of different technologies that make up the family of AI, machine learning (ML) is the most basic form. ML consists of teaching computers to perform tasks with a specific goal by using examples instead of explicit programming [3,4,5]. Although we may not know it by its actual name, ML is already part of our everyday life. It is the system used by Netflix to recommend which film we may want to watch; by our email applications to filter spam; or even by our car to assist us in driving and parking [6]. Deep learning (DL) is a subset of ML, which does not require the direct intervention of humans as it learns how to make predictions or decisions through complex algorithms known as artificial neural networks. This makes it practically impossible to understand how the DL model arrives at the answers it provides [7]. DL is particularly useful in the recognition of images and of speech [8]. More complex forms of AI include natural language processing (NLP) models, which are able to comprehend human language and/or unstructured text and transform it into structured data [5], and large language models (LLMs) and NLP tools that can understand and generate human-like language [8]. Both NLP and LLM models learn from vast amounts of data in a self-supervised manner.

ChatGPT is an example of an LLM, which has been trained on data obtained from open sources on the internet [9,10,11]. Although ChatGPT was not developed to provide medical knowledge, other systems such as Google’s Med-PaLM 2 have been specifically trained on medical data for such purposes [12,13].

Considering how the body of medical knowledge required to treat a patient has exponentially grown in the last forty years (while, in 1980, it doubled every seven years, in 2010, the doubling time was less than 75 days), access to a comprehensive and readily available source of such knowledge may support clinical teams, ensuring all patients receive the best possible care [5].

The possible applications of AI to medicine are not limited to providing updated and accessible medical knowledge. ML and NLP technologies may, in fact, help in the screening of clinical and radiological diagnoses; in the choice of medical treatments and in tailoring patients’ management; in surgery; and in the reduction of the administrative burden [9,14,15]. Moreover, when applied to research settings, AI technologies may significantly alleviate the burden of recruitment, monitoring, and insertion of data in clinical trials [16,17].

At the present moment, most applications of AI to health care have been validated in prospective clinical studies with proven diagnostic performance and safety and are yet to be validated in randomized controlled trials [5]. As such, their use is still mostly hypothetical or limited to research settings. However, this form of technology is rapidly evolving and is expected to soon overflow in all medical fields, including gynecology, where some tools are already present [18]. Given this background, we aimed to summarize the available evidence regarding the possible application of AI to the management of women with endometriosis and adenomyosis (Table 1). Whether we like it or not, medicine, like many other aspects of our lives, has taken this direction and it is our responsibility as physicians and researchers to build upon our knowledge in this regard. AI could actually represent the missing element which may help us overcome the limitations in both endometriosis and adenomyosis clinical management and research that human brains have not yet been able to master.

## 2. Methods

We considered all original studies analyzing the possible role of AI in the management of endometriosis or adenomyosis. Articles were included if they were written in English and were published in peer-reviewed journals. No limit to the year of publication or to the patients’ age was applied. Abstracts, conference papers and articles not reporting original data were excluded.

Articles were identified by a PubMed search that was carried out on 11th December 2023 using the keywords “endometriosis” and “adenomyosis” in combination with “artificial intelligence”, “AI”, ”machine learning”, “deep learning”, “Chat GPT”, “Med-PaLM 2”, “large language models”, and “natural language processing”. References from relevant publications were also screened and further articles were searched using PubMed’s “Similar articles” and “Cited by” functions.

## 3. Applications of AI to the Clinical Management of Endometriosis and Adenomyosis

### 3.1. Role in the Formulation of Clinical Diagnoses

Artificial intelligence technologies may represent a novel opportunity to decrease the diagnostic delay that notoriously characterizes both endometriosis and adenomyosis [19]. In fact, by providing updated medical knowledge, accessible both to physicians and to the public, they may ensure clinicians have considered all differential diagnoses that explain patients’ symptoms, as well as increasing patients’ awareness of the disease, deconstructing the common belief that menses are painful by definition [20]. On the one hand, it is of uttermost importance that algorithms are accurate and characterized by high specificity in order to minimize the risk of generating inappropriate referrals and unjustified anxiety. On the other hand, AI technology should be further trained to sort through the myriad of symptoms and information a patient may report, deducing and selecting only the most relevant.

The minimization of pain is a major determinant of diagnostic delay in endometriosis and adenomyosis, and is especially common among teenagers and individuals with superficial endometriosis [21,22]. In the former case, minimization often arises from a low awareness of the disease in young women among clinicians and among patients and their families [23]. In the latter, both insufficient medical knowledge and radiological challenges (superficial endometriosis is not visible by the means of an ultrasonographic examination) play a role [22]. In both cases, AI may represent a game changer.

By encouraging women’s referral to specialists, and by increasing physicians’ medical knowledge, AI may help redefine the epidemiology of endometriosis and adenomyosis. This may be particularly true for adenomyosis, which is rarely diagnosed in adolescents, although it actually appears to be more frequent in this specific population than commonly believed [24].

The accuracy of chatbots in resolving medical cases and in providing accurate medical knowledge has been proven in various studies, although their performance is still far from that expected in clinical settings. When submitting a series of diagnostically challenging medical cases to ChatGPT4, Kanjee and colleagues found that the chatbot gave the right answer in 39% of cases, and included the correct answer among differential diagnoses in 64% of cases [25]. When questioned with the United States Medical Licensing Examination quiz, chatbots with no specialized training passed or nearly passed the test [26]. Astonishingly, in another study comparing the performance of ChatGPT4 with that of medical journal readers in resolving real-life medical cases, the chatbot correctly diagnosed 57% of cases, compared to 36% correct diagnoses given by the journal readers [27]. It must be pointed out that, in the latter study, the population of human journal readers was poorly characterized and that their level of medical skills, as well as the effort they put into answering questions, was unknown. As to what specifically regards gynecology, Ozgor and Simavi recently analyzed the accuracy of ChatGPT in answering questions about endometriosis. As many as 91% of questions were answered accurately, although among the questions based on the ESHRE endometriosis guidelines [28], accuracy was lower (67.5%) [29]. Considering the speed at which LLM technologies are expanding, and, consequently, how fast they may improve in preciseness and accuracy, these results are encouraging.

As well as representing a source of medical knowledge, AI may provide algorithms for the prediction of the likelihood of endometriosis in patients with chronic pelvic pain or with infertility. This will be made possible by the fact AI is able to detect patterns in large volumes of data. Such algorithms will have to be validated on large populations from multiple centers in order for their data to be applicable on a vast scale.

Heterogeneous approaches have been suggested by various study groups to build such algorithms, most of which achieved sensitivity and specificity above 85%. Data used so far to build prediction or diagnostic models include the following: clinical features (age, presence and severity of symptoms, comorbidities, infertility, previous surgery) [30,31,32,33,34]; serum and salivary biomarkers [35,36,37,38]; genomics, transcriptomics, metabolomics, proteomics and methylomics data [39,40,41,42,43,44]; lipidomic data from endometrial fluid [45]; gene, mRNA and proteomic and transcriptomic expression in the endometrium [46,47,48]; mixed data [49,50]; and radiologic images [51,52]. However, the majority of these studies, which have been comprehensively analyzed in Sivajohan and co-workers’ recent review [19], were retrospective, meaning that the models were trained and validated on patient datasets, rather than in vivo on humans. Moreover, the efficacy of AI in predicting and/or diagnosing endometriosis and adenomyosis was not compared with that of existing decision algorithms and of clinical diagnostic tools. Further research is needed in this regard.

### 3.2. Role in the Formulation of Radiological Diagnoses

As to what specifically concerns the radiological diagnosis of endometriosis and adenomyosis, AI has a great potential to improve its quality by learning to detect anomalies in ultrasonographic and MRI images. This is made possible by the fact AI is able to match imaging findings with previously registered data [2,53,54].

Computer-assisted interpretation of radiological data is already in use in other medical fields and, in some cases, appears to be as accurate as experienced radiologists [55,56]. Moreover, DL methods may improve diagnostic accuracy by eliminating subjectivity, and may provide diagnoses in a few seconds [53]. Various studies have been published on this topic so far; however, most algorithms are lacking adequate validation and generalizability and are currently limited to research purposes [57,58,59].

In the management of endometriosis and adenomyosis, this kind of technology may be of particular aid in distinguishing patients with and without the disease, especially in complex cases with atypical presentations or in settings in which expert radiologists or ultrasound examiners are not available. In their recent pilot study on 50 individuals with a surgical diagnosis of endometriosis, and an equal number of individuals with at least one symptom of endometriosis but without a diagnosis of endometriosis, Balica and co-workers used five different DL methods to aid the sonographic diagnosis of the disease. AI-assisted diagnosis was feasible and efficacious and was able to predict endometriosis with a 90% AUC and 80% accuracy [60].

AI may also prove useful in the differential diagnosis of endometriosis from other benign conditions. In Hu and colleagues’ retrospective study, DL was used to distinguish ovarian endometriosis from tubo-ovarian abscesses (TOAs) on ultrasonographic images [57]. Like endometriomas, TOAs may, in fact, present as hypoechoic avascular cystic masses within the context of a pelvis distorted by adhesions. Astonishingly, when comparing AI’s performance with that of three ultrasound examiners and of the plasma concentrations of carbohydrate antigen 125 (CA 125), DL’s performance was superior [57].

A particular aspect of the ultrasonographic diagnosis of endometriosis is represented by the evaluation of the Pouch of Douglas (POD), which may be obliterated by adhesions between the retrocervix, the anterior wall of the rectum and the uterosacral ligaments [61]. POD obliteration is particularly important to recognize pre-operatively as it increases surgical complexity and the risk of complications, as well as plays a role in the disease’s prognosis [62,63,64]. A sonographic marker of PD obliteration has been described (“sliding sign”); however, it relies on the operator’s expertise and largely depends on inter-observer variability [65,66]. To overcome such limitations, Maicas and colleagues analyzed the ability of DL in defining the state of the POD by classifying transvaginal ultrasound videos depicting positive and negative “sliding signs”. The accuracy, sensitivity and specificity of such a model were all just short of 90%, indicating high diagnostic performance [64].

However, not all studies on this topic have confirmed these encouraging results. When comparing DL’s ability to recognize adenomyosis on uterine ultrasonographic images with that of intermediate skilled trainees, Raimondo and co-workers found that the trainees’ accuracy was higher than DL’s (70% versus 51%). However, DL’s specificity, i.e., the ability to correctly identify healthy uteruses, was higher than the trainees’ (82% versus 69%). The authors concluded that the DL model could prove useful in limiting the over-diagnosis of adenomyosis, although the literature appears to suggest we are facing the opposite problem (under-diagnosis), especially in some categories of patients, first and foremost adolescents [67].

### 3.3. Role in the Choice of Medical Treatments and in the Customized Management of Patients

It has been estimated that the third leading cause of death in the United States is represented by medical errors [68]. Fortunately, human error in medical practice usually leads to less serious consequences than death; however, even mistakes with less catastrophic consequences require attention.

Errors are an inevitable limitation of human actions, which may arise from distraction, work overload, or lack of knowledge. Prescribing medical treatments for which an individual presents contraindications, overlooking the presence of the interactions between medications in patients with comorbidities, or simply not choosing the most adequate molecule for a given patient are extremely frequent events [68]. Integrating human activity with AI-driven control systems may represent an innovative solution to mitigate the frequency of such errors and limit their consequences in clinical practice.

Monophasic low-dose hormonal contraceptives and progestins are considered first-line options for the treatment of endometriosis, as they have the most favorable safety/efficacy/tolerability/cost profile [28,69]. As to what regards adenomyosis, no guidelines have been approved at the present time, although levonorgestrel-releasing intrauterine devices (LNG-IUDs) appear to be an effective first-line treatment [70,71,72]. Oral progestins, dienogest, in particular, and combined oral contraceptives (COCs) have also been proven to be effective in these patients [73,74,75]. Second-line treatment consists of GnRH analogues, both for endometriosis and for adenomyosis [73]. Surgery is the only therapeutic option in specific cases of endometriosis, including obstructive uropathy; bowel occlusion or subocclusion; ovarian cysts with a diameter greater than 5 cm or suspected of malignancy; and cases in which hormonal therapies are not tolerated or contraindicated [28]. Conversely, for women with adenomyosis, especially during childbearing age, surgery is rarely an option, and is usually limited to hysterectomy in patients in perimenopause [73].

However, no size fits all. Patients’ age, ongoing treatments for other conditions, comorbidities, response to treatment and life plans all strongly influence the choice of medical and surgical treatment. AI may aid in facilitating such choice, starting from the choice of prescribing treatment at all. In fact, not all patients with endometriosis or adenomyosis are promptly prescribed adequate therapy. This applies especially to adolescents, for whom hormonal treatment does not appear to be the standard of care [74], although it could prove particularly beneficial in improving painful symptoms and in reducing the risk of disease progression [70].

The World Health Organization has provided recommendations for contraceptive use in women with medical conditions or medically relevant characteristics, which should be routinely applied by gynecologists to avoid drug–disease interactions. Some of these recommendations are routinely addressed in medical practice, although not all are. Moreover, authors suggest that within the same class of molecules, some hormonal therapies are more adequate than others for specific populations.

AI algorithms based on such recommendations and on the most recent literature may guide physicians in an accurate manner towards the adoption of customized therapies, also providing alerts for drug–drug interactions. For example, AI algorithms may advise and remind clinicians to avoid molecules which have a greater effect on bone mass density (BMD) (dienogest monotherapies, GnRH analogues) in younger women; those with greater androgenic effects (NETA) in women with hyperlipidemia, hypercholesterolemia, or signs of hyperandrogenism; and those associated with a higher risk of thromboembolic events (COCs containing third- and fourth-generation progestins, COCs with ≥ 30 mcg ethinyl estradiol, transdermal patches, vaginal rings) in women with known risk factors. Conversely, they may suggest the adoption of therapies which have none or minor adverse effects on BMD (LNG-IUD, continuous use of COCs, estrogen–progestin transdermal patches, vaginal rings) in adolescents or in women with known risk factors for osteoporosis; those which are less likely to induce adverse serum lipid changes (COCs containing micronized 17β-estradiol (E2), or E2 valerate, or estetrol) in women with hyperlipidemia or hypercholesterolemia; those associated with a reduced risk of venous thromboembolisms (second-generation progestins, LNG-IUD, subdermal implant progestins, COCs containing micronized 17β-E2, or E2 valerate, or estetrol) in those with known risk factors for cardiovascular accidents; and those which are approved as contraceptives (COCs, LNG-IUDs, desogestrel monotherapies and etonogestrel subdermal implants) in women desiring contraception [70,75,76,77,78,79,80,81,82].

AI technologies may not only assist clinicians in the choice of the most adequate treatment for a given patient, but they may also guide clinical decisions by predicting outcomes such as reproductive prognosis and cancer risk. Knowledge regarding their reproductive prognosis may empower patients, enabling them to adjust their life projects around their condition and increasing their perception of being taken care of, ultimately improving their satisfaction and their adherence to treatment [22,83,84]. Risk models helping clinicians predict which patients are more likely to encounter a malignant transformation of endometriosis may help identify who requires a timely surgical treatment [85]. Chao and co-workers recently developed a risk model through ML that can predict the risk of endometriosis-associated ovarian cancer, with sensitivity and specificity both short of 90%. The model was built using clinical characteristics including, among others, age, age at menopause and size of the ovarian cysts. Although it was created within a pilot study, and certainly requires further validation, it is a promising example of how AI may facilitate the early identification of malignant transformation, helping clinicians recognize those patients in which risk-reducing medical or surgical interventions should be carried out [85].

Back in 2005, Awaysheh and co-workers reviewed 97 studies analyzing the effects of computer-based clinical decision support systems in various medical fields (cardiology, general surgery and psychiatrics) and found that such systems improved practitioners’ performance in 64% of studies, as well as improved patients’ outcomes in 13% of studies. Although dated, this study provides encouraging results which can only be improved by the advancing application of AI technology [86].

### 3.4. Role in Surgical Treatment

Surgery may also benefit from AI, especially through its integration with robotic surgery systems. In fact, DL may assist surgeons by providing real-time guidance in the interpretation of anatomy and in the individuation of endometrial foci (by matching robotic surgery images with pre-operatory imaging data). By combining such information with surgeons’ movements, it may warn operators of the risk of complications at an early stage, and, by comparing surgeons’ movements with the data acquired from expert surgeons, it may also be of aid in training. Moreover, by memorizing such movements, it may convert the entire process to an automated surgery. In fact, AI systems can be programmed to cut tissue or suture with a high degree of precision [2,87,88].

At the present moment, the literature regarding the application of DL to endometriosis and adenomyosis surgery is limited to Hernández and colleagues’ 2022 publication [88]. In their observational study, the authors used DL algorithms to quantify the level of indiocyanine green, and consequently of blood perfusion, in bowel anastomoses during laparoscopic bowel resection for endometriosis. The model obtained 92% accuracy and may represent a first step towards the adoption of AI in surgical settings to increase precision and reduce the frequency of complications [89].

### 3.5. Role in Reducing the Burden Linked to Administrative Work

The tight link between productivity pressure and burnout is clear. It has been estimated that physicians now spend more than 50% of their time updating electronic health records. Since the advent of COVID-19, they have also been spending an increasing amount of time in their out-of-office hours taking care of the exorbitant volume of electronic communication with patients [22,90,91]. This comes at the expense of efficacious communication, empathy, and clinicians’ psychological wellbeing [22], which all affect patients’ empowerment, satisfaction with treatment, adherence to treatment, symptom perception, and ability to remain integrated in society despite suffering from a chronic condition [92,93,94]. However, the solution may be round the corner. In fact, AI-powered technologies may not only be of aid in providing summaries of large medical records, filtering and drafting medical notes and e-mails, generating laboratory and prescription orders, cataloguing diseases according to their ICD, and scheduling appointments; they may do so at a greater speed than humans and with greater accuracy [10,58]. This would enable physicians to have more time actively interacting with their patients, while reducing working hours, and ultimately reducing their risk of burnout [95]. Machines can actually augment our humanity [12] and this has been proven in Ayers and colleagues’ study, in which ChatGPT was found to respond with higher quality and more empathetic answers to patients’ health care questions compared to physicians [13].

Further administrative work, which may be taken over by AI, includes the management of staff rotations and that of operating room slots. In the latter case, optimizing slots by using systems which are able to predict operating room use time may considerably decrease waiting lists, improving health quality at a national level [5]. Also, by creating large datasets including electronic health records from all medical institutions in a given country, AI may be of aid in the establishment of the transition from a fee-for-service reimbursement model to a value-based care model. In fact, by comparing the indication of medical and surgical treatments with available protocols and guidelines, AI technologies may help identify and reduce low-value health care, where value is considered as the relation between the potential benefits, harms and costs of a given medical intervention [96,97].

## 4. Applications of AI to Endometriosis and Adenomyosis Research

As soon as privacy issues are resolved, AI technologies may significantly alleviate the burden of patient recruitment, site monitoring and the insertion of data in clinical trials by engaging both with participants and with researchers in chat, audio or avatar modes. From a practical point of view, by being able to adapt to any language and communication style, AI models may interact with patients, answering their questions about recruitment and checking they respond to inclusion criteria, while scheduling appointments and collecting data. As to what regards researchers’ tasks, AI technologies may be of aid in analyzing data, producing reports, interpreting results and searching the literature. This may not only lessen researchers’ workloads, but also considerably reduce the current global expenditures on clinical trials, which are estimated to be more than $50 billion [17,98].

In endometriosis and adenomyosis research, much effort has been made over the past decades to better understand disease pathophysiology. Despite numerous genetics and public health studies being carried out, there have been no breakthroughs in the understanding of these conditions. However, these kinds of studies, which are typically based on a vast amount of data, may be facilitated in the future by the use of AI-based technologies, which are able to classify highly complex data and identify patterns and associations while adjusting for co-exposures [99]. Authors have already started adopting ML for such purposes. In their epidemiological study, Matta and co-workers analyzed the presence of pollutants in the abdominal tissue of 55 women with deep endometriosis and 144 healthy controls [99]. ML models showed a high classification performance and were able to consistently reveal a number of pollutants associated with the disease. Genetic studies analyzing the expression of long non-coding RNAs [100] and of genes [101,102] reported an up-regulation of the genes involved in vasculogenesis, cell proliferation, cell–matrix adhesion, cell differentiation, extra-cellular matrix, remodeling muscle contraction, apoptosis, immune response and chemotaxis.

Electronic medical records are an underutilized source of longitudinal data which could be extracted and analyzed using AI technologies to aid such a field of research [103], along with data obtained from symptom-tracking apps.

As the transition to LLM-augmented medical research advances, the World Association of Medical Editors (WAMEs) has recently alerted editors regarding the possible downfalls of ChatGPT-generated medical writing, including the risk of plagiarism and of false content [104]. WAME recommendations included the following points: chatbots should not be considered as authors, as they cannot take responsibility for their paper; authors should disclose their use of chatbots and provide detailed information on how AI technologies were used; authors should be held responsible for contributions given to their manuscripts by chatbots; and editors should have access to updated tools to recognize AI-generated content [104]. Accordingly, medical journals have started publishing their own guidelines regarding the use of chatbots in medical writing [105].

## 5. Limitations and Challenges of AI

So far, we have enumerated the numerous ways in which AI may overcome human limitations and, as such, facilitate the work of physicians and clinicians and improve patients’ experience. However, AI also entails limitations and, as such, needs to be strictly regulated in order for it to be used in a safe manner, with respect to ethics and human rights. In this regard, while appraising AI’s potential to improve healthcare worldwide, the World Health Organization has outlined six areas for an efficacious regulation of AI, which all governments should adhere to [106]. Such areas include the following: protecting human autonomy (humans should remain in control of medical decisions and patients should understand the role AI plays in their care); promoting human wellbeing and safety (AI should not result in mental or physical harm to individuals); ensuring transparency and intelligibility (AI technologies should be understandable to developers, users and regulators); fostering responsibility (regulatory principles should be applied to the algorithms); ensuring equity (AI should be accessible worldwide, not only in high-income settings, and AI should not encode biases which may marginalize minority groups); and promoting responsiveness and sustainability (AI should respond to requirements and should be consistent with global efforts to reduce humans’ impact on the environment) [106].

Although it is unlikely physicians’ autonomy in making decisions for their patients will be taken over by AI, patients using LLMs as medical resources may be misled by receiving incorrect information. In fact, LLMs’ human-like conversational style and their apparently authoritative and plausible answers to human prompts easily gain users’ trust [16,107]. This poses safety risks, as AI technologies are surely promising, yet still far from being infallible, especially when answering medical queries [9]. On some occasions, they even make information up, as has been found when asking a chatbot to provide references for its answers [107]. In Goodman and colleagues’ cross-sectional study on the accuracy of chatbot-generated answers to questions based on medical guidelines, the percentage of completely incorrect answers was as high as 8% [9]. This is due to the fact chatbots are not conscious; they simply rearrange existing data which has been previously input in the system, without being able to discriminate between reliable and unreliable sources [98]. LLMs specifically trained on medical guidelines and on the most recent literature are being developed to overcome such pitfalls [108,109]. Moreover, at the present moment, AI technology is still not able to account for certain real-world aspects of making a clinical diagnosis or solving a medical case such as sorting through the myriad of symptoms and information a patient may report and deducing the most salient, as well as understanding nuances linked to the patient’s emotional wellbeing [58]. Such drawbacks of AI may have more serious repercussions on patients’ health if clinicians who use these technologies in their clinical practice are not aware of their limitations, and, as such, do not verify their veracity [5]. Understanding how different AI methods work, and what their competencies and their limitations are, is fundamental to safeguard individuals’ safety. It is the health workers’ duty to keep up to date in this regard to protect both their patients and themselves [19].

As international medical organizations and journals have already started doing, it is essential that regulatory principles for AI application in healthcare are defined at a global level. Guidance is needed as to what regards patient privacy, the protection of sensitive data [14,110] and malpractice liability [111]. Considering that AI applications’ efficiency depends on the quality of the data they are fed, it is urgent to find a balance between protecting sensitive data and feeding AI systems with the most vast and complete amount of data possible. Also, currently, no LLM has been reviewed by the US Food and Drug Administration, nor by its equivalents worldwide. As such, if a patient experiences malpractice as a result of the use of AI-based technology, no one may be liable for the patient’s injury except the clinician, who should always verify AI’s outputs before relying on its use [111].

A further aspect of AI, which will need to be brought to the attention of users, is the risk of introducing, in AI applications, systemic biases which result in the marginalization of minority groups [112]. Training AI models with data that are not representative *in toto* of the population the algorithm is going to be used on is the basis of such biases [113]. In a recent study exploring the use of an AI model to aid in the interpretation of chest radiographs, the authors noted that, although trained on datasets of thousands of images, the AI model under-diagnosed diseases in underserved and minority groups [114]. This may be the case not only of ethnic minority groups, but of all of those individuals whose data were too scanty or difficult to retrieve for the AI models to be trained on. Access to the internet and to AI facilities is another aspect which should be considered when analyzing the risk of the marginalization of underserved communities.

## 6. Conclusions

Although human intellect has brought about an astonishing improvement of life expectancy and of the quality of life in the last few centuries, human interventions are intrinsically prone to error. Errors may limit the quality of care and the ability to further improve both clinical research and medical practice. This is where AI may fit in; while it may not be able to reduce human errors to zero, it can reduce them significantly.

Despite the fact most AI applications have not been subject to randomized controlled clinical trials, their potential in improving healthcare is evident. In the management of endometriosis and adenomyosis, AI technology may not only improve patients’ experience by contributing to a reduction of diagnostic delay, guaranteeing a more efficient patient–clinician interaction, enhancing surgical outcomes, and facilitating customized forms of care, but it may also improve physicians’ experience by decreasing the administrative workload, aiding diagnoses and the choice of medical treatments, increasing surgeons’ expertise and facilitating clinical research. At a global level, AI may improve health services’ efficiency and speed up the transition from a fee-for-service reimbursement model to a value-based care mode. Such improvements may occur through AI’s ability to carry out automation, doing so with greater precision and in a fraction of the time compared to humans.

More research is needed before such technologies can be implemented in clinical practice and strict regulation is necessary before these algorithms are introduced in everyday practice. In the meantime, it is our standing as clinicians and researchers to expand our knowledge in this regard, overcoming our skeptical view of machines as usurpers ad recognizing that this form of man-made technology may actually be the solution to overcome men’s intrinsic limitations.

## Figures and Tables

**Table 1 jcm-13-02950-t001:** Possible applications of AI to the management of endometriosis and adenomyosis.

	Possible Role of AI	Consequences on Disease Management
Formulation of clinical diagnoses	Providing updated medical knowledge accessible to physicians and to the public	It may aid differential diagnosis It may increase patients’ awareness of the disease It may redefine epidemiology
	Providing algorithms for the prediction of the likelihood of endometriosis in women with CPP or infertility	It may decrease diagnostic delay It may improve clinical outcomes
Formulation of radiological diagnoses	Detection of anomalies in US and MRI images and performance of diagnoses in a few seconds	It may improve efficiency It may assist diagnosis in complex cases or in settings in which specialists are not available
Choice of medical treatments and customized patient management	Recognizing drug-disease and drug–drug interactions	It may reduce the risk of prescription errors
	Prediction of reproductive prognosis and cancer risk	It may improve patient empowerment It may help identify high-risk patients
Surgical treatment	Overlapping robotic surgical evaluations with pre-operatory imaging data	It may assist in the interpretation of anatomy and in lesion detection
	Comparing surgeons’ movements with those of experts	It may be of aid in training
	Warning of the risk of complications	It may improve surgical outcomes
Administrative work	Providing summaries of medical records, filtering and drafting notes and e-mails, generating prescription orders, cataloguing diseases according to their ICD at a greater speed than humans and with greater accuracy	It may reduce physicians’ risk of burnout It may increase patients’ satisfaction
	Managing operation room slots and scheduling appointments	It may decrease waiting lists
	Creating large datasets of electronic health records from all medical institutions	It may help in the transition from a fee-to-service reimbursement model to a value-based care model
Endometriosis and adenomyosis research	Engaging with participants and with researchers in chat, audio or avatar modes.	It may alleviate the burden of patient recruitment, site monitoring and insertion of data
	Analyzing vast amounts of data, classifying highly complex data, identifying patterns, producing reports and searching the literature	It may reduce the expenditure on clinical trials It may help improve the understanding of disease pathophysiology

Table legend: CPP: chronic pelvic pain; US: ultrasonographic; MRI: magnetic resonance imaging; ICD: International Classification of Diseases.

## Data Availability

Not applicable.

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
