# Peer review of "Artificial Intelligence in the Management of Women with Endometriosis and Adenomyosis: Can Machines Ever Be Worse Than Humans?"

_jcm, 2024, doi:10.3390/jcm13102950_

Round 1

Reviewer 1 Report

Comments and Suggestions for Authors

Dear author’s

I was pleased to review your article and i have the following comments:

First of all in the introduction it is mandatory to estabish the aim of your study.

 The subject is not new. I think that your paper should be change in editorial.

I suggest you to explain how the information was obtained.

The article is very narrative and it is very difficult to assess the mail question of your paper.

Please explain the novelty or what new info brings your paper to the existing literature.

Minor English edits.

Author Response

We greatly appreciate the opportunity to revise and improve the above-referenced manuscript. We thank the reviewers for their useful suggestions provided. Our revisions are outlined below, as well as our responses to each of the issues addressed by the reviewer. All revisions are specified and explained in this text, and highlighted in the revised manuscript. Page and line numbers are indicated where the modifications have been made.

Reviewer 1

  1. Dear author’s, I was pleased to review your article and i have the following comments:

First of all in the introduction it is mandatory to estabish the aim of your study.

Authors’ response (AR): we have reported in the introduction what the aim of the review was: “Given this background, we aimed to summarize the available evidence regarding the possible application of AI to the management of women with endometriosis and adenomyosis”.

  1. The subject is not new. I think that your paper should be change in editorial.

AR: we understand the reviewer’s point of view, however the section editor specifically asked for a review and in our opinion our manuscript can be considered as such.

  1. I suggest you to explain how the information was obtained.

AR: we have introduced a Methods section to better explain how the information was obtained.

  1. The article is very narrative and it is very difficult to assess the mail question of your paper. Please explain the novelty or what new info brings your paper to the existing literature.

AR: the main question of our paper is indeed vast. We believe the novelty of this manuscript is in the fact that it comprehensively summarizes the available evidence regarding the possible use of AI in endometriosis management, giving the reader an overview of the state of the art. It is very narrative but this was our aim, as we wanted to introduce and accompany the reader in a new dimension (i.e. AI), stimulating his/her curiosity towards a subject which may not be well known. Our impression is that physicians are still far from being experts in this field, which they might consider unknown and not important for their everyday work. This is the reason why we decided to adopt a style which is not technical or difficult to comprehend, but rather accessible to any gynecologist who wishes to start widening his/her knowledge on the topic.

  1. Minor English edits.

AR: we have reviewed the manuscript for typos or for grammar mistakes.

Reviewer 2 Report

Comments and Suggestions for Authors

I would like to express my gratitude to the authors for their effort in compiling this well-structured manuscript. However, I find it lacking a clear purpose or conclusion intended for the readers.

Author Response

We greatly appreciate the opportunity to revise and improve the above-referenced manuscript. We thank the reviewers for their useful suggestions provided. Our revisions are outlined below, as well as our responses to each of the issues addressed by the reviewer. All revisions are specified and explained in this text, and highlighted in the revised manuscript. Page and line numbers are indicated where the modifications have been made.

Reviewer 2

  1. I would like to express my gratitude to the authors for their effort in compiling this well-structured manuscript. However, I find it lacking a clear purpose or conclusion intended for the readers.

AR: we wish to thank the reviewer for his comment. As reported in our answer to reviewer n. 1’s fourth question, the purpose of our review is to comprehensively summarize the available evidence regarding the possible use of AI in endometriosis management, giving the reader an overview of the state of the art. It is very narrative but this was our aim, as we wanted to introduce and accompany the reader in a new dimension (i.e. AI), stimulating his/her curiosity towards a subject which may not be at all well known. The impression is that physicians are still far from being experts in this field, which they might consider unknown and not important for their everyday work. This is the reason why we decided to adopt a style which is not technical or difficult to comprehend, but rather accessible to any gynecologist who wishes to start widening his/her knowledge on the topic.

Reviewer 3 Report

Comments and Suggestions for Authors

As a gynecologist and informaticist, this reviewer can be considered qualified to review this manuscript.

We are all aware of the potential value of AI in many aspects of our life, including Medicine, but we must refrain from using comments which only suggests what is possible, and sometimes inflating its value, without considering the objectively derived facts which we seek to do in academic debates. While the promise to “improve the quality of care” for endometriosis is a laudable goal, yet this manuscript fails to objectively compare the results of using ML for the diagnosis with any human involvement in its diagnosis and management. The question of whether AI can improve the care for endometriosis compared to humans, was not answered. Yet the suggestion to abandon “magical thoughts” when discussing AI is exactly what is represented here.

The diagnostic use of Three-Dimensional sonography for its role in the diagnosis of endometriosis along with obtaining the history and performing an examination should certainly be utilized, though nowhere was the value of AI demonstrated to augment this function. While the use of algorithms to improve upon human abilities, in order to provide clinical decision support, may certainly be desirable. However, this manuscript failed to compare its value with the current standard of care. Recognizing the need to understand the proper nuances in the diagnosis of endometriosis is certainly important, as was described, but it is the inclusion of AI elements in this diagnostic process that was not convincingly identified in this manuscript.

Author Response

We greatly appreciate the opportunity to revise and improve the above-referenced manuscript. We thank the reviewers for their useful suggestions provided. Our revisions are outlined below, as well as our responses to each of the issues addressed by the reviewer. All revisions are specified and explained in this text, and highlighted in the revised manuscript. Page and line numbers are indicated where the modifications have been made.

  1. As a gynecologist and informaticist, this reviewer can be considered qualified to review this manuscript. We are all aware of the potential value of AI in many aspects of our life, including Medicine, but we must refrain from using comments which only suggests what is possible, and sometimes inflating its value, without considering the objectively derived facts which we seek to do in academic debates. While the promise to “improve the quality of care” for endometriosis is a laudable goal, yet this manuscript fails to objectively compare the results of using ML for the diagnosis with any human involvement in its diagnosis and management. The question of whether AI can improve the care for endometriosis compared to humans, was not answered. Yet the suggestion to abandon “magical thoughts” when discussing AI is exactly what is represented here. The diagnostic use of Three-Dimensional sonography for its role in the diagnosis of endometriosis along with obtaining the history and performing an examination should certainly be utilized, though nowhere was the value of AI demonstrated to augment this function. While the use of algorithms to improve upon human abilities, in order to provide clinical decision support, may certainly be desirable. However, this manuscript failed to compare its value with the current standard of care. Recognizing the need to understand the proper nuances in the diagnosis of endometriosis is certainly important, as was described, but it is the inclusion of AI elements in this diagnostic process that was not convincingly identified in this manuscript.

AR: we agree with the reviewer that objectivity is an essential aspect of science. For this reason we did not aim to convince or promise readers that AI will surely improve quality of care. Rather, we suggested that this could happen, as the advancement of AI-related technology suggests that medicine of the near future could gain from such a kind of aid. However, it is not possible to state this as a certainty as (as written in the manuscript) randomized controlled trials comparing AI-aided diagnostics with human involvement are not yet available and as such their use is still mostly hypothetical or limited to research settings.

Unfortunately, at the present time it is not possible to answer the question as to weather AI can improve medical care as evidence in this regard is extremely scarce and does not allow to compare AI with the standard of care.

Moreover, to our knowledge original studies specifically regarding the use of AI-augmented 3D ultrasonographic diagnosis of endometriosis and/or adenomyosis are not available.

Reviewer 4 Report

Comments and Suggestions for Authors

This is a narrative review of the applications of AI in the diagnosis and management of endometriosis and adenomyosis. I have a few comments to make:

 Line 54: “complex task, which would typically involve biological brainpower”: please specify whether this definition includes physical tasks, which may be facilitated via autonomous (or semi-autonomous) surgical robots, whereby biological movement would also be substituted.

Line 83: “The possible applications of AI to medicine are not limited to providing updated and accessible medical knowledge (Table 1).” You mention applications in Medicine in general, while the Table is about applications in Endometriosis/adenomyosis in particular.

Section 1: This section as a whole is too long and includes superfluous information on the definition of AI, along with examples. I would recommend reducing it in length, focusing on the application in Medicine.

Section 2: since you present distinct areas for AI application, you could consider presenting them individually, for example talking about applications in clinical diagnosis in a separate section from radiological diagnosis.

Section 2.1: While the authors pose some interesting prospects for the future, they provide few actual data. This is understandable to some extent, given the novelty of the topic. I would advise shortening this section to include the most relevant information. Additionally, the authors should mention how AI would address the issue of correctly interpreting and inferring type and severity of symptoms from the patients. Answering some clinical questions that provide clear, unambiguous clinical data and instructions is one thing but correctly interpreting a patient’s description of their symptoms, while focusing on the most important ones is a different matter altogether. You do mention it in the end, but I think that you should mention it here as well.

Line 95: What do you mean 90% probability? This is not a commonly used term in neither diagnostic accuracy nor diagnostic concordance studies.

Comments on the Quality of English Language

English is appropriate for the most part, I only detected some minor issues that could be solved via careful proofreading.

Author Response

We greatly appreciate the opportunity to revise and improve the above-referenced manuscript. We thank the reviewers for their useful suggestions provided. Our revisions are outlined below, as well as our responses to each of the issues addressed by the reviewer. All revisions are specified and explained in this text, and highlighted in the revised manuscript. Page and line numbers are indicated where the modifications have been made.

Reviewer 4

  1. This is a narrative review of the applications of AI in the diagnosis and management of endometriosis and adenomyosis. I have a few comments to make:

Line 54: “complex task, which would typically involve biological brainpower”: please specify whether this definition includes physical tasks, which may be facilitated via autonomous (or semi-autonomous) surgical robots, whereby biological movement would also be substituted.

AR: we have corrected the text as suggested, including physical tasks among AI’s complex tasks.

  1. Line 83: “The possible applications of AI to medicine are not limited to providing updated and accessible medical knowledge (Table 1).” You mention applications in Medicine in general, while the Table is about applications in Endometriosis/adenomyosis in particular.

AR: we have moved the reference to Table 1, as suggested.

  1. Section 1: This section as a whole is too long and includes superfluous information on the definition of AI, along with examples. I would recommend reducing it in length, focusing on the application in Medicine.

AR: we have reduced the length of section 1 (section 2 in the revised manuscript) as suggested by the reviewer. We decided to keep the short reference to the applications of AI to everyday life as, given the narrative style of the manuscript, we deemed it important for the reader to be able to associate AI to known daily experiences.

  1. Section 2: since you present distinct areas for AI application, you could consider presenting them individually, for example talking about applications in clinical diagnosis in a separate section from radiological diagnosis.

AR: we have separated the paragraph concerning clinical diagnosis from that regarding radiological diagnosis, in section 2 (section 3 in the revised manuscript), as suggested.

  1. Section 2.1: While the authors pose some interesting prospects for the future, they provide few actual data. This is understandable to some extent, given the novelty of the topic. I would advise shortening this section to include the most relevant information. Additionally, the authors should mention how AI would address the issue of correctly interpreting and inferring type and severity of symptoms from the patients. Answering some clinical questions that provide clear, unambiguous clinical data and instructions is one thing but correctly interpreting a patient’s description of their symptoms, while focusing on the most important ones is a different matter altogether. You do mention it in the end, but I think that you should mention it here as well.

AR: we have modified section 2.1 (3.1 in the revised manuscript) according to the reviewer’s suggestions. We have shortened the section to include only the most relevant information and added a short paragraph regarding the need for AI technology to discriminate between important and less important information.

  1. Line 95: What do you mean 90% probability? This is not a commonly used term in neither diagnostic accuracy nor diagnostic concordance studies.

AR: the terminology we used was the same Balica and co-workers used in their study. We have changed the term probability with AUC (area under the curve).

  1. English is appropriate for the most part, I only detected some minor issues that could be solved via careful proofreading.

AR: we have proof-read the manuscript according to the reviewer’s suggestion.

Round 2

Reviewer 1 Report

Comments and Suggestions for Authors Dear author’s    I already tell you that the article is more likely an editorial.  The clinical utility is not very clear explain.  I didn’t understand what you want to tell with this. This is the reason why we decided to adopt a style which is not technical or difficult to comprehend, but rather accessible to any gynecologist who wishes to start widening his/her knowledge on the topic.  

Reviewer 2 Report

Comments and Suggestions for Authors

As I also deal scientifically with AI in the medical context, I assumed that a similar level of knowledge would be widespread. However, you are right that not everyone has the same level of knowledge and not everyone is equally familiar with AI. In this sense, the article could draw the attention of medical colleagues to the possibilities of artificial intelligence.

Reviewer 3 Report

Comments and Suggestions for Authors

The issues previously mentioned have all been adequately addressed.

Reviewer 4 Report

Comments and Suggestions for Authors

My comments for the first round of reviewing have been adequately addressed and I thank the authors for their work.

I would personally recommend that the Introduction (section 1) is further shortned, but I understand the opinion of the authors about wanting to maintain some daily applications of AI for the readers. Therefore this is only a suggestion and not a critical revisions that needs to be made.

Regarding my 5th comment from the previous round about the questionable ability of AI to discrn and organise patient symptoms, I am pleased that the relevant information has been provided by the authors. However, I believe that you should also mention it in lines 156-170, where the specific examples of chatGPT's ability to answer structured questions with accuracy. Patients' descriptions are frequently lacking (patients may not be able to provide precise descriptions of location, intensity and frequency of symptoms), compared to structured questions that provide all relevant information in an easy to discern way. Therefore, the findings of the studies you present should be interpreted carefully, as the do not necessarily mean that the same efficiency and accuracy will be maintained in a real clinical setting.

All my other concerns have been addressed very well and I have no further comments.